# Reasons for non-adherence to cardiometabolic medications, and acceptability of an interactive voice response intervention in patients with hypertension and type 2 diabetes in primary care: a qualitative study

Aikaterini Kassavou, Stephen Sutton

► Prepublication history and additional material are available. To view these files please visit the journal online (http://dx.doi.org/10.1136/bmjopen-2016-015597).

Behavioural Science Group, Department of Public Health and Primary Care, The Primary Care Unit, University of Cambridge, Cambridge, UK

**Correspondence to**
Dr Aikaterini Kassavou; kk532@medschl.cam.ac.uk

## ABSTRACT

**Objectives** This study explored the reasons for patients' non-adherence to cardiometabolic medications, and tested the acceptability of the interactive voice response (IVR) as a way to address these reasons, and support patients, between primary care consultations.

**Design, method, participants and setting** The study included face-to-face interviews with 19 patients with hypertension and/or type 2 diabetes mellitus, selected from primary care databases, and presumed to be non-adherent. Thirteen of these patients pretested elements of the IVR intervention few months later, using a think-aloud protocol. Five practice nurses were interviewed. Data were analysed using multiperspective, and longitudinal thematic analysis.

**Results** Negative beliefs about taking medications, the complexity of prescribed medication regimens, and the limited ability to cope with the underlying affective state, within challenging contexts, were mentioned as important reasons for non-adherence. Nurses reported time constraints to address each patient's different reasons for non-adherence, and limited efficacy to support patients, between primary care consultations. Patients gave positive experiential feedback about the IVR messages as a way to support them take their medicines, and provided recommendations for intervention content and delivery mode. Specifically, they liked the voice delivering the messages and the voice recognition software. For intervention content, they preferred messages that were tailored, and included messages with 'information about health consequences', 'action plans', or simple reminders for performing the behaviour.

**Conclusions** Patients with hypertension and/or type 2 diabetes, and practice nurses, suggested messages tailored to each patient's reasons for non-adherence. Participants recommended IVR as an acceptable platform to support adherence to cardiometabolic medications between primary care consultations. Future studies could usefully test the acceptability, and feasibility, of tailored IVR interventions to support medication adherence, as an adjunct to primary care.

### Strengths and limitations of this study

► Participants: patients presumed to be non-adherent to cardiometabolic medications, and nurses with experience in advising patients for medication taking, recruited by primary care practices within different levels of deprivation.
► Analysis: in-depth, multiperspective, thematic analysis of patients and nurses' views on the reasons for non-adherence; current actions to address these reasons; and the acceptability of interactive voice response (IVR) delivery mode, and content, to support patients' adherence between primary care consultations.
► Design: face-to-face interviews to explore reasons for non-adherence, and think-aloud protocol to explore the acceptability of the IVR.
► Applicability: consideration should be given when translating these results to different populations and settings.

## INTRODUCTION

Primary care consultations aim to advise and support patients in their decision to take the prescribed medications[1]; however, a substantial proportion of patients cease, or do not take, their medications as prescribed, after leaving the consultation room.[2 3] Non-adherence to medications contributes to low treatment efficacy and increased National Health Service (NHS) costs,[4] particularly for people who use healthcare services regularly, such as patients with either type 2 diabetes mellitus (T2DM), or hypertension, or those with both health conditions.[2 4 5]

Patients' non-adherence is influenced by one, or both, intentional (INA) and non-intentional (NINA) reasons.[6 7] INA refers to those reasons that are under patients' conscious

decisions to miss, or alter, medication regimens; whereas NINA refers to those reasons that are outside patients' conscious consideration of their behaviour. Currently, we do not know the reasons for non-adherence, and how to effectively address them[8]; and practitioners' time is limited and expensive. Both NICE (National Institute for Health and Care Excellence) guidelines[1] and a report commissioned by the Department of Health,[4] recommend the development, and test, of novel interventions to actively involve patients in addressing non-adherence, not only during, but also between primary care consultations.

Interventions using existing telephone systems are promising platforms to support patients, as an adjunct to primary care.[9 10] Our recent systematic review found that automated telephone-based interventions can double the odds to promote medication adherence, when added or compared to usual care.[11] Such interventions enable behaviour change using automated, and interactive, voice response (IVR) messages. In the UK, adults use 216.8 billion minutes in voice calls (142.8 billion min via mobile phones and 74 billion min via fixed lines) every year.[12 13] Despite the major potential to reach a large number of patients, even in remote areas, and facilitate the delivery of low-cost medication adherence services,[14] the acceptability of such systems has not been tested in the UK primary care setting.

To address this gap, this study aimed to develop, and assess the acceptability of, elements of an IVR intervention to support adherence to cardiometabolic medications. To achieve this, the present study identified the reasons for patients' non-adherence; the current actions to facilitate adherence; and obtained views and recommendations about an IVR intervention to influence these reasons and support adherence, as an adjunct to primary care consultations.

## METHODS
### Recruitment
The study was approved by the NHS Ethics Committee (REC reference: 14/LO/1872), and the NHS Cambridgeshire and Peterborough Clinical Commissioning Group (CCG), before any participant was approached. Primary care practices within the CCG, with the highest recorded numbers of patients with hypertension and T2DM, and at different levels of deprivation, were eligible for this study. Sixty-one practices were invited by the research team, and five of them participated in the study, of which four had list sizes smaller than the average, and one larger than the average, when compared to other practices within the CCG. The practices varied in deprivation level. Three practices invited nurses, and four invited patients, to take part in the study.

Nurses eligible for inclusion were those who were involved in advising, encouraging, or supporting patients to adhere to their treatments. Eligible nurses were approached face-to-face by the practice manager, given

the study information pack, and invited into the study. Five nurses, from three practices, provided written informed consent, and took part in the interview. Patients eligible for inclusion were those: (a) aged 40 and over; (b) with a primary diagnosis of hypertension and/or T2DM; (c) who were non-adherent to their medication regimens, as indicated by clinical measures closest to the last refilled medication prescription, or with gaps in ordering or refilling repeat prescriptions during the last three months; and (d) had visited the practice at least once during the last six months.

The practice manager selected eligible patients from the practice database. A GP screened the list of eligible patients for the main exclusion criteria (e.g., recent acute health event; diagnosis of dementia, aphasia, or other cognitive difficulties; severely impaired hearing or speaking). Practice staff posted study invitation packs to those selected for inclusion. In total, 150 invitations were sent to eligible patients, and 19 expressed an interest, and provided written informed consent to participate in the study. The recruitment was completed when saturation of the data had been achieved.[15] The study recruited less nurses than stated in the protocol, because the collected data had reached saturation, after implementing constant comparisons between themes.

### Interviews
At baseline, face-to-face interviews were conducted using a semi-structured guide to identify the reasons for non-adherence. The interview guide was informed by a review of evidence and theory-based research.[16 17] The theoretical concepts that best explained medication adherence change, were mapped onto interview questions, and informed the interview schedule (see online supplementary files 1 and 2). Interview prompts facilitated the in-depth exploration of meanings and the identification of themes.[18] We asked nurses their views on: (a) patients' experiences with taking medicines, and barriers to adherence; (b) their current actions to address non-adherence; and (c) their recommendations about the development of a telephone-based intervention, as an adjunct to primary care. Interviews with nurses were conducted from January 2015 until April 2015. We asked patients about their views on: (a) their condition; (b) medication adherence; (c) barriers and facilitators in taking their medicines; and (d) whether, and how, an automated telephone-based intervention could support them take their medications as prescribed. Baseline interviews with patients were conducted from July 2015 untill September 2015.

Based on the baseline interviews, an IVR platform to deliver the intervention messages was developed.[9] The content of the IVR messages was developed by combining the findings from the baseline interviews and systematic literature reviews, which explored the effective behaviour change techniques (BCTs) to support medication adherence.[10 11 19] At the follow-up interviews, a think-aloud protocol to gain participants' experiential feedback on elements of the IVR was used. Participants were asked to

trigger and complete one or two IVR calls, and verbalise their thoughts during and after the calls. Each IVR call lasted approximately 1–5 min. Participants were asked and prompted to think aloud their views on the delivery mode, and the content of the messages;and asked their recommendations on how the IVR messages could be improved, and support them to adhere to their medications best. Follow-up interviews with patients were conducted from March 2016 until April 2016.

The first author constructed the interview guide, think-aloud protocol, IVR messages, and conducted interviews and analysis, with input from the second author. The first author had experience and training in qualitative research. The second author is also a leading expert in the topic of research. Patients were interviewed at their homes (n=11 baseline and n=12 at follow-up), at the University (n=7 at baseline), or at their workplace (n=1 at baseline and n=1 at follow-up). There was the intention for all the follow-up interviews to take place at participants' homes, to facilitate the collection of contextual information.[20] All nurses were interviewed at their practices. All interviews were face-to-face with only the researcher and the interviewee present, apart from one baseline interview, where the interviewee's partner was present to assist with translation. Each baseline interview lasted on average one hour, and each follow-up interview 30 minutes. All interviews were digitally recorded, and transcribed by an independent service. Transcripts were double-checked against audio recording and field notes for accuracy,by the researcher.

### Analysis
The data collected from the two cross-sectional sets of interviews (i.e, baseline interview with patients and nurses) were analysed using in-depth, multiperspective, thematic analysis.[21] All interviews were analysed separately. After that, data were integrated into broader descriptive categories. For baseline interviews, latent themes from patients and nurses' interviews were identified inductively and synthesised into one analysis, capturing both perspectives. Themes from the follow-up interviews were mapped onto those from baseline interviews, and any new themes (e.g., on acceptability of the IVR delivery mode) were treated separately.[22]

### RESULTS
### Participants
In total, 37 interviews were conducted: 19 patients were interviewed at baseline, 13 of whom were interviewed few months later; and 5 nurses. Patients' mean age at baseline was 62.3 years, seven of them were female, 13 reported a primary diagnosis of hypertension, three reported T2DM, and the rest reported diagnosis of comorbidities of cardiovascular conditions (e.g., stroke) and T2DM. More information about the practices,nurses, and patients' characteristics can be found in table 1.

### Reasons for medication non-adherence
Participants reported that they do not initiate taking their medications as prescribed, when they do not understand the need to take medicines to support them manage their long-term health conditions.

*"I think probably they don't think they need them, to be honest with you. 'I don't need them what are you giving me these for?' yeah 'I'm well, look at me, I've been like this all me life'."* Patient B2

*'The trouble with diabetes is people don't realise that they've got it, and they go on for a couple of years before it's discovered, and they probably have gone all this time and not felt ill at all; and then they've been given a tablet, so they haven't really had a time when they felt ill.'* Patient B4

*"I suppose, deep down they are not really happy to be taking medication, and they think if they ignore the problem then, or they keep the tablets on the side ,and they think 'oh I'll take it, if I really need it'."* Nurse A5

Some participants reported that they intentionally changed or stopped the course of their medications, when they had no health benefits by taking them. The underlying affective state, associated with taking medications, was important to mobilise sustained adherence. Participants, who reported not feeling any different by taking medications, seemed to report low motivation to adhere to their prescriptions.

*'He say that he takes one tablet one day, two tablets, second tablet second day, and it helps, but after, for example, the second tablet on the fourth day it doen't help any more ….[he became] accustomed with this tablet, with the medications [and don't want to take more]. He went more times to the GP and they gave him the same. The tablets did nothing.'* Patient B17

*'I don't really ever consider that I've got high blood pressure. I just don't think about it… with blood pressure it's just how I feel, I don't know what it feels like. It is, like, say, there's nothing really to tell me day-to-day.'* Patient B14

*'I often think all these six going down my throat ,all in one go, and they're all releasing their selves, but I don't feel any different no…You don't realise what these tiny little things have got in them do you?'* Patient B4

*'If they're feeling well, they'll drop it out or, drop down… they think they don't need it. They just don't want to take something every day.'* Nurse A4

Participants, who had unpleasant side effects from taking medications, reported stopping or skipping their tablets.

*'The side effects there were which made me feel ill, there's no point in taking pills that make you feel ill, is there?'* Patient B2

*'I felt as though I was walking through treacle, I wasn't right at all, so no, we don't take that one anymore.'* Patient B10

**Table 1** Practice and participants' characteristics.

| Practice Deprivation level within CCG, area, practice list size on average within CCG, patient/doctor ratio, patient/staff ratio (excluding doctors) | Participants | | | | | |
|---|---|---|---|---|---|---|
| | | Patients | | | | |
| | Nurses' pseudonyms and gender | Pseudonyms | Age | Gender | Primary diagnosis | Participated at follow-up interview* |
| Average deprived; urban; small; 1136.6/1; 516.6/1 | A1 female | B1 | 80 | Male | T2DM and hypertension | Yes |
| | | B2 | 70 | Male | Hypertension and stroke | Yes |
| | | B3 | 67 | Male | Hypertension | Yes |
| | | B4 | 70 | Female | T2DM | Not interested |
| | | B5 | 66 | Male | Hypertension | Not interested |
| | | B6 | 71 | Female | Hypertension | Yes |
| Least deprived; rural; small; 1208/1; 302/1 | A3 female | B7 | 57 | Female | Hypertension | Moved area |
| | A4 female | B8 | 61 | Female | Hypertension | Yes |
| Less deprived; rural; large; 1275.37/1; 351.8/1 | | B9 | 51 | Female | Hypertension | Yes |
| | | B10 | 64 | Female | Hypertension | Yes |
| | | B11 | 70 | Male | Hypertension and T2DM | Yes |
| | | B12 | 64 | Male | Hypertension | Yes |
| | | B13 | 65 | Male | Hypertension | Yes |
| | | B14 | 44 | Male | Hypertension | No time |
| | | B15 | 52 | Male | Hypertension | Yes |
| | | B16 | 61 | Male | Hypertension and T2DM | Yes |
| Most deprived; urban; small; 702.6/1; 301/1 | A2 female | B17 | 45 | Male | Hypertension | Excluded |
| | | B18 | 50 | Female | T2DM | Yes |
| Least deprived; rural; small; 1378/1; 393.8/1 | A5 female | B19 | 77 | Male | T2DM | No time |
| | | Participants' mean age in years | 62.3 | | | |

Average list size: CCG n=933,265 of whom n=117,784 patients with hypertension and n=43,296 with diabetes.
*Note.* CCG, Clinical Commissioning Group; A(x) letter code to anonymise nurses (number code to anonymise nurses); B(y) letter code to anonymise patients (number code to anonymise patients); T2DM, type 2 diabetes mellitus;*Yes, if participants took part in the interviews; or reported reasons for not taking part.
, ; .

*'I'll come across people, who won't take something the doctor has prescribed, because they're getting side effects that they don't like. For example, Metformin which gives them tummy ache, and the doctor might have prescribed say, two, or three doses a day, and they'll only take one…generally speaking, Metformin in the first couple of weeks can cause a lot of indigestion, nausea, tummy aches, that kind of thing,and often patients don't like it, so they stop taking it.'* Nurse A5

The regimens people were prescribed were also mentioned as an important barrier for medication non-adherence. People taking lots of pills reported limited ability to cope with multiple regimens, and those taking few pills reported lack of motivation to continue taking their tablets as prescribed.

*"Sometimes I think it's that they, they've got so much medication, they say 'I've got so much medication, I forget'."* Nurse A3

*"If I'm routinely asked to take a pill without really knowing why I'm taking it, it's not a very good incentive to take that pill…they may walk out of this surgery thinking 'do I really need to take this [medication]?', I lose the prescription or not quite make it to the chemist."* Patient B13

Some participants reported beliefs that behaviours can have an accumulative or counteractive effect to each other. In these cases, participants tended to alter their dosages according to the other behaviours, enacted at times and/or places, related to their mediation taking

behaviour. For example, people reported that they intentionally alter, or miss, their tablets when socialising, especially when their social environment did not support them taking their medications.

*'The only time I don't take it, if I go to a birthday party, or wedding, or whatever, where you abuse yourself with a bit too much alcohol. I think I'd rather not take the tablets, because I don't want it interfering with the alcohol in me, there's enough poison in there as it is.'* Patient B11

*'Yeah, I won't take it if I'm out. If we go out for dinner, I won't take my tablets, because I don't want other people to see me taking them.'* Patient B18

Some participants also reported difficulties in taking their medicines, when they change their everyday routine and/or they are busy. For example, one participant reported forgetting taking his pills, when going away for the weekend; whereas another reported forgetting, when he has multiple activities to do in a day. Additionally, nurses reported participants' memory issues as a reason for NINA.

*'I just take them when I get up, and if I get up in a rush, then that is when I can forget. 'I will take the cardinipine later in the day', but I've forgotten after about lunchtime.'* Patient B8

*'Oh well I got down to me daughter's in Kent once, when was that? Not last Christmas, the Christmas before, and I'd forgotten. I'd taken most of me pills but I'd forgotten half of them, so we had to go round to the local doctors and get a prescription, luckily I'd got a copy of my prescription what I take.'* Patient B1

*'Memory problems, so we would see maybe forgetfulness, living on their own, nobody to prompt them.'* Nurse A1

*'I came across a patient the other day who admitted that he took it when he remembered, and that was possibly two, three times a week.'* Nurse A5

### Current actions to promote medication adherence and recommendations for an intervention

Participants reported that one way to remember to take their medications as prescribed was to add reminders to their environment. These cues were already associated with activities of their daily routine, and were enacted at close time and space proximity with taking medications.

*'[In the morning] I have a coffee, I put the mug of milk [for the coffee] in the microwave…before I go to bed, I put the pills in the microwave…to make me remember and get into the habit of it, 'you don't have your coffee until you've had your medication, otherwise you don't have a coffee'. You must associate that and put them together, I do and it works.'* Patient B3

*'I suppose it's quite a little routine isn't it? I actually put a handkerchief by the bed, so that I wake up, either when the alarm goes, or usually I wake up half an hour or so before the alarm, and I find it and take it with a shot of water, I have a bottle of water by the bed.'* Patient B5

*'I think it's about patterns in my life, so every morning I have a tray, that's where my phone, my keys are, everything I need for the next day I put on that tray…so every morning I'd put that [tablet] on my little tray, so really just finding somewhere every day where I've got the things that I need for the next day has been a help to me. Because when I did not have that pattern, I'd go to work and 'oh I need my pill now, it's at home.'* Patient B14

Patients were satisfied with the current practices in primary care, in terms of providing them with advice on how to take their medications. Some nurses also reported that they encourage patients to add cues to their environment, to help them cope with forgetfulness. Another nurse reported that she advises patients, about how to cope with their underlying affective state during the course of their medications.

*"Just a lot of encouragement, there was a lot of contact, and checking 'have you done this' and 'did you do that' or 'well done you', trying to give them a bit of a boost. I think that's about the best you can do."* Patient B10

*"I said to her 'would you go out the house without cleaning your teeth', 'no', 'so put it by your toothbrush'."* Nurse A1

*'The other thing I tell them is that 'how you think you feel is often not a good indication of actually what state you are in, because you just get used to feeling in a certain way'.'* Nurse A3

However, both patients and nurses reported challenges with the time available within a consultation to address each patient's needs regarding their medications. Moreover, participants reported challenges with their abilities to support themselves (in the case of patients), or provide support to patients (in the case of nurses), to adhere to prescriptions daily, and/or during conflicting conditions.

*'I think, if you've been advised to take it, and you don't, what can they do? I don't see what help they can do.'* Patient B6

*'I had a new patient's appointment on Tuesday afternoon, and she's got a list of things that she's got to do, she's got to go through what medication you're on, she's got to test your urine, she's got to take your blood pressure, and ask these questions, and she's got 10 min to do it.'* Patient B9

*"Yeah on the prescriptions it will say 'take one tablet daily or every day'… people don't want to sit and read it."* Nurse A1

*"They don't actually understand how the stuff works in the first place so but, but I know that I've had patients back who are saying 'I didn't know that' and I know that I told them two years ago, because it's documented in their notes, so they don't always listen either, that's the other thing… But we really haven't got time to be going into everybody's medications, and why, and how they take them, I just think you can't go round people's houses and ram their pills down their throat twice a day, can you?."* Nurse A4

*"Sometimes patients say things to you and then you find out, when you actually check their medications or prescriptions,*

*that they're not doing what you asked them to do really. . and I suppose there's always this feeling of 'I'll do something about it', very motivated when they leave the surgery, 'I'll sort it out, I am nogoing to have medicines, I'm going to sort this problem out' but actually when it comes to it, it's very difficult to change some lifestyle."* Nurse A5

Both nurses and patients expressed favourable views on an automated, telephone-based intervention; and suggested that such an intervention could be more effective, if it was tailored to each patient's reasons for not taking medicines as prescribed.

*'If it was set automatically, that would be brilliant… as long as it was automatic and didn't cost any money, for the health service, not just me.'* Patient B8

*'It's like there's not a pill that will sort everyone out. It's a very complicated thing, you really need to fit the answer to the person, so you need to find out from them… in order to get someone to take their medication you've got to understand how they fit their world together…so, if someone doesn't establish routines, you've got to find out how they're keeping their life together.'* Patient B5

*'One size doesn't fit all really does it? Everybody's different and what works for one person doesn't always work for another…ask them what they would prefer, I think that could be good.'* Patient B9

*'I think it would need to be different for different people, I don't think you could do that across the board.'* Nurse A5

### Acceptability of the IVR delivery mode

IVR seemed to be an acceptable platform to deliver messages for medication adherencefrom the majority of the participants. Specifically, participants liked the voice delivering the call, the navigation options, and the voice recognition software. Even one person, who was reluctant about the automatic element of the call, reported that receiving an automated call would have been better than forgetting medications.

*'Now that you've told me it's not a person I'm talking to, I can say what I like, can't I? A lot of people when they pick up that phone are going to think they are talking to a real person. They're not. It's a nice, it's well pitched, the lady is a pleasant voiced lady, it doesn't sound like a cold call or anything…[voice recognition] it's better than pressing buttons, I hate pressing buttons it's awful.'* Patient B13

*"I don't know how to explain it, but to be actual talk to a person than a machine is entirely different …that might put certain people off, but on the other hand, it might think 'thank goodness that came through, I'll take my medication'."* Patient B6

*'So many firms if you press this and you press that, press one, press two, press three. I: Would you prefer that, or would you prefer to speak to (an automated voice response system) R: Yeah this is better, that's much better.'* Patient B8

Participants seemed to also like the interactive element of the IVR delivery mode, which they used to report

missing some of their dosages, and/or to ask for advice on how to cope with non-adherence.

*"If you've got something wrong with you, then you probably wait a few months to go to the doctors…or 'll think 'oh I can get over this' and so on, but really in the background in their mind they are slightly worried and they might want a bit more information. I: So do you think they would do that through that system? R: I think they would, as a starting point, because it's not going, it's not talking on a one-to-one with somebody, you're actually just accessing an accurate database which is reliable."* Patient B16

*'I did forget to take my tablets this morning, can I still take them now or should I just wait for the next day?'* Patient B18 inbound call

*"if I ring up now and say 'oh my prescription's run out can I order a new prescription?'"* Patient B12

### Acceptability of the intervention content

Most participants expressed favourable views about the content of voice messages, which included BCTs, to address reasons for non-adherence. Specifically, they found useful the messages that included 'information about the health consequences' of adhering, or not, to their medicines, especially since hypertension and/or T2DM can be asymptomatic. Specifically, participants highlighted the importance of messages that reminded them of the reasons they need to take their medicines as prescribed. Participants reported that receiving messages at occasions, which they usually find challenging to adhere to their medications, could support them take their tablets as prescribed.

*'R: people get it into their heads that they feel alright and therefore they don't need to take it. I thought it was good in the message that you said that they need to take it regularly in order to maintain, to keep their blood pressure down.'*

*'I: do you think this information is not known to people?'*

*'R: maybe they choose to ignore it. They probably do, but high blood pressure is one of those things that unless you actually feel it you're not aware that it's a problem.'* Patient B9

Additionally, most participants liked simple messages that prompted them to take their medications at a due time, and facilitated conscious thinking of their behaviour.

*"'Hello Mrs, have you taken your drugs today?' just to remind them."* Patient B2

*"I'm thinking if they ring up, it's like some of my medication, like the warfarin, I have to take it at a certain time, if I don't take it then I've basically got to wait 24 hours and take the next. If it says 'have you taken your medication?' you say 'no' then that comes across and say 'oh best I'd take it'."* Patient B12

*"I got a call at midday saying 'have you taken your medications today?'…it would cause me to think before I answered [laughs]."* Patient B16

However, messages that included information about the duration of their health condition, rather than information about how to support themselves to adhere to their medications; and messages that asked them to put effort in self-monitoring their medication taking behaviour (e.g., keeping a diary every day) seemed not to be acceptable.

"*It's saying to me, 'you've been taking this medication for seven years'. I don't need them to tell me why, like I know really, I just need reminding to take it. The other thing is, it says about writing down in your diary, but if I was getting the calls I would be like 'what are you talking about?' do you know what I mean? I wouldn't want to be told to write it down. I think if I was that organised that I would write down in a diary, I wouldn't need somebody to ring me. I literally would just want it to say 'take your medicine'.*" Patient B18

Personalised voice messages (i.e., that included patients' names) seemed to be acceptable to the majority of the participants.

'*I think they will because it says your name, so you're waiting for it.*' Patient B18

'*It feels like somebody who's closer to you, somebody's who's on, familiar with you, so in that respect it's better.*' Patient B9

## DISCUSSION
### Principal findings
Beliefs about the need to take medications, the number of prescribed doses, and the ability to cope with the underlying affective state, within challenging contexts, were mentioned as important reasons for adherence to cardiometabolic medications. Nurses reported time constraints to address each patient's different reasons for non-adherence, and limited efficacy to support them between, and sometimes during, the primary care consultations. Both patients and nurses had favourable views on an automated, telephone-based intervention, and recommended messages to be tailored to each patient's needs for taking medications. Patients expressed positive experiential feedback about elements of the IVR, and provided recommendations for intervention content and delivery mode. Specifically, participants liked the voice delivering the messages and the voice recognition software. For intervention content, patients expressed a preference for messages that are tailored to their conditions, include 'information about health consequences', or simple reminders of the behaviour. The interactive element of the interventions seemed to be most useful for people to request support, when they had missed their tablets. Including elements of personalisation would increase uptake and initial engagement with the intervention.

### Strengths and limitations
This study invited patients whose practice records suggested that they were not refilling their prescriptions and/or not managing their blood pressure or glucose levels. Insights obtained from these patients can usefully inform the content, and delivery, of a medication adherence intervention within primary care. However, consideration should be given when translating these results to different populations (e.g., patients who do not attend practices or those not participating in research), and other settings (e.g., community).

### Comparison with existing literature
Previous studies have investigated the reasons for medication non-adherence,[23] as well as patients or practitioners' views of an intervention to address non-adherence.[24 25] In line with previous research, participants' attitudes about the need to take medications, especially when they are prescribed complex medication regimens,[26] seem to be important reason for non-adherence. Similarly, previous intervention development, and feasibility, studies suggested that automated, telephone-based interventions should remind people about their prescription plan, provide patients with the options of interactive communication, and tailored intervention content.[27 28] However, none of these studies had investigated the underlying reasons that influence non-adherence to cardiometabolic medications (i.e., underlying affective state); whether, and how, current practices address these reasons; or considered multiple views and recommendations on intervention delivery mode and content to support adherence. Most importantly, none of the previous studies have investigated the acceptability of the IVR to address the reasons of non-adherence in the UK primary care setting, which is an important element when developing novel interventions.[29]

### Implications for research and practice
IVR seems to be an acceptable mode to influence the determinants of medication non-adherence in patients with hypertension and/or T2DM. This study suggests that within the time constraints, and limited resources of the primary care, practitioners could briefly assess patients' beliefs about taking medications, ability to cope within challenging conditions, and the complexity of prescriptions; and then signpost them to an IVR adherence intervention. The results of this study also suggest that reminders of the behaviour, and 'information about health consequences,' are acceptable techniques to address reasons for non-adherence.

### Unanswered questions and future research
Future studies aiming to develop such interventions in primary care could usefully tailor the intervention to each patient's INA and NINA reasons. Future studies could include effective and acceptable BCTs to address these reasonsand promote adherence. Future studies could integrate participants' views on barriers and facilitators

to uptake, retention, and engagement, with such interventions. Participants' views and recommendations on the delivery mode, such as the clarity, tone, volume, intensity, speed rate, rhythm, and vocal expression of the messages; the navigation options; and the frequency and duration of the calls, should be further explored. Additionally, future studies could explore participants' views, satisfaction, mastery, and actions upon the content of IVR interventions; and obtain recommendations for improvement. Future studies could also explore whether these, or additional components proposed by theories such the COM-B model, could impact on medication adherence,[30] using rigorous designs.

**Contributors** AK and SS designed the study. AK conducted all tasks involved in completing this study, with substantial input from SS.

**Funding** This report is independent research by the National Institute for Health Research. The views expressed in this publication are those of the author(s) and not necessarily those of the NHS, the National Institute for Health Research or the Department of Health.

**Disclaimer** This report is independent research by the National Institute for Health Research. The views expressed in this publication are those of the author(s) and not necessarily those of the NHS, the National Institute for Health Research or the Department of Health.

**Competing interests** None declared.

**Patient consent** Obtained.

**Provenance and peer review** Not commissioned; externally peer reviewed.

**Data sharing statement** Documents describing the process of data analysis are available by request to the first author.

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
