## [Reviewer comments · BMJ Open]

ARTICLE DETAILS

TITLE (PROVISIONAL)	Reasons for non-adherence to cardio-metabolic medication, and acceptability of an interactive voice response intervention in patients with hypertension and type 2 diabetes in primary care: a qualitative study
AUTHORS	Kassavou, Katerina; Sutton, Stephen

VERSION 1 - REVIEW

REVIEWER	Hizlinda Tohid Department of Family Medicine, Universiti Kebangsaan Malaysia Medical Centre, Malaysia
REVIEW RETURNED	09-Jan-2017

GENERAL COMMENTS	For me, qualitative studies that explore perception on IVR intervention aimed at improving medication adherence among those who are not compliant to their medications are significant and important. However, this article was not well written to give significant impact to readers. This mainly because the article fails to emphasise the focus of the study. The authors stated three main objectives which I think best to be answered by three different methodologies. Due to lack of focus, the findings were superficial reflecting poor exploration of the issues understudied (not in-depth enough). Thus, the study appears less meaningful. The methodology of the study was appropriate for developing an IVR intervention. Even though it is important to explore reasons for non-adherence to medications and strategies used to improve adherence, it should not be the main objective of the study. If exploring reasons for non-adherence to medication is the main objective of the study, the methodology should ensure obtaining maximum variation of opinion and views through maximum variation sampling. This can be achieved by including heterogeneous participants with important characteristics e.g. those with intentional versus unintentional non-adherence to medications and those with various severity of non-adherence (occasional versus persistent) etc. If exploring strategies that can improve medication adherence is the focus of the study, interviewing multiple key informants (e.g. patients who are currently/used to not compliant to their medications, doctors, nurses, pharmacists or family members/carers) is very important. Even though the authors claimed that they carried out multiple perspective interviews, interviewing only patients and nurses may not be enough to ensure triangulation. Generally, the methodology used was not able to produce rich and in-depth findings that can meet all objectives. In my opinion, exploring participants' perception on the IVR intervention developed to improve medication adherence among patients with hypertension
--

and/or diabetes should be the focus of the article. Therefore, this article requires major corrections.

Specific comments

1. Literature review

- Since some readers may not be familiar with IVR intervention, it would be better if the authors could provide some overview about IVR i.e. definition and how it works.
- Unfortunately, the authors only mentioned about telecommunication intervention in general. It is not clear how IVR intervention could improve medication adherence. Is it through provision of cues/reminders or advice/tips for behavioural change or both?
- Justification of the study was not clear. The gap of the knowledge was not emphasised. The authors did mention that telecommunication interventions could improve patients' adherence to medication but such interventions have not been developed and tested in the UK. How this gap in practice relates to the study? What is the connection between not having such intervention and the need to explore patients' reasons for non-adherence and perception on strategies that can improve adherence (including IVR intervention)?

2. Study design

- I think it is alright for not specifying the research design according to the qualitative research tradition as long as the methodology is clearly described, particularly related to sampling of participants, theoretical perspectives, process of data collection and data analysis. The description can help readers to determine appropriateness of the methodology used in answering the objective(s) of the study.

3. Sampling of the participants

- As I mentioned before, it must be dependent on the objective(s) of the study.
- It is good that the authors described the various settings where the participants were from (i.e. primary care practices with different level of deprivation, list size and urban/rural). However, it is better if the authors could specify the average list size of the practice and the patient: doctor ratio or patient: staff ratio. This information could reflect the work burden of the practices which could prevent their health providers from giving intensive management for patients' non-adherence. It will be great if the authors could include information about availability of special program to address medication non-adherence.
- Seeking opinion from multiple sources about the IVR intervention is important by triangulating sources of data. But in this case, I think, involvement of nurses is unnecessary if the focus of the study is to explore patients' opinion regarding the IVR intervention. The nurses' involvement could not really be considered as having multiple perspectives as there was no triangulation.
- Purposeful sampling of the patients were not clearly described. Instead of specifying the exclusion criteria, the reasons why they were sampled should be explained. Their specific characteristics should be emphasised. Selecting patients merely because they were not compliant to

	medication is not enough. It should be guided by other factors as well.  • Random sampling of the eligible patients as described in the text (page 6, line 41) is irrelevant in qualitative study. • The impact of interviewing a participant with language barrier (page 7, line 47-48) should be discussed as well. • Although only 13 out of 19 patients were interviewed after experiencing the IVR intervention, the point when data saturation occurred should be explained. The authors must be aware that data saturation can easily be achieved if the participants are homogenous, preventing rich data to be obtained. 4. Theoretical framework/perspectives  • Theoretical framework/ perspectives of the study was poorly explained. The authors mentioned about previous research (including stage theories – reference 15) that informed the development of the semi-structured interview guide. Behavioural Change Techniques (reference 9) was also utilised in the development of the IVR intervention. However the theories and perspectives were not elaborated. • Apart from development of the interview guide and the IVR intervention, did the authors use them for other purposes (e.g. sampling of the participants and analysis of the data)? 5. Data collection  • Data collection period was not specified. • The semi-structured interview guides used before and after exposure to the IVR intervention should be included. • It is good that the authors described the venues of the interviews. However, they did not need to state their intention to carry out the interviews at the participants' home (page 7, line 43-44). As long as the place of interviews are chosen by the participants and they are comfortable and conducive for recordings. • Double checking of the transcription is very important to ensure trustworthiness of the study. Usually researchers double-check transcripts against voice recordings to ensure accuracy of the transcription, but not against field notes as stated by the authors (page 7, line 51-52). Field notes are used to describe the settings, participants and behaviour of the participants which are not captured by the recordings. Researchers could also write the gist of what are said by the participants and their reflection of the interviews, initial interpretation or working hypotheses in the field notes. 6. Description of the IVR intervention  • Details about the IVR intervention such as the flow of the automated IVR call, questions, option of answers and advice given should be described. This could help readers to understand the participants' opinion. • Did the first author use her voice to record the messages? If so, could the participants identify her voice? How this method could influence the findings? 7. Ethical issues  • The authors should specifically include information about informed consent from the participants.
--	--

	8. Data analysis • Method of analysis was poorly described. The authors claimed that the data was analysed inductively using thematic analysis. However, it was not clear how the themes/ subthemes/ categories were formed. For me, the subthemes/ categories were not apparent. It appears that the authors formed the main themes according to the objectives of the study i.e. reasons for non-adherence, actions to promote medication adherence and acceptability of the IVR intervention. The analysis was mainly descriptive and not much abstraction was done.• It was not clear whether data collection and data analysis were concurrent.• I do not understand why the authors mapped the themes from the follow-up interviews against the baseline interviews. This is because the issues explored during the baseline and follow-up interviews were different except for the participants' opinion on the IVR intervention. Yet, there was no findings that report the difference in their opinion. 9. Interpretation of data • The findings appear superficial. This may be because exploration of each issue was not in-depth enough or the article lacks focus.• I feel that some of the findings were not consistent with the data/ verbatim. For example, the authors interpreted Quote 9 as favourable views on the intervention but for me, the verbatim indicates unfavourable views since the participants highlighted the complexity to personalise the intervention to patients' individual needs. Verbatims by patient B6 in Quote 10 and patient B18 in Quote 15 seem to support this unfavourable views as well.• I do not understand with the statement in page 11, line 15-16: 'Moreover, participants reported challenges with their abilities to support themselves..'. What are the challenges? It appears that the verbatim did not reflect the statement.• I also feel that the interpretation of the verbatim by patient B16 in Quote 14 was incongruent. I do not agree that the patient preferred simple messages as he/ she queried about the question which was non-specific and vague. Since the patient might be taking more than once daily medications, asking him/ her: "Have you taken your medication today?" at midday could be confusing and difficult to answer. Was it referring to the morning dose or afternoon/ night dose? The data could also emphasise the importance of the timing of the call. Perhaps calls to remind adherence of the morning dose medication should be done in the morning. 10. Discussion • I feel that the discussion of the findings was superficial too and not critical enough. There was no new knowledge, perspective or insight synthesised through comparing the study's findings with previous studies. For example, the issue related to the need for personalised intervention was not discussed. Since a number of participants were quite sceptical about developing personalised IVR intervention, the issue should be further elaborated. Is it really possible to personalise the intervention? Was there similar intervention that managed to do so?• Other issues that need to be discussed include:
--	---

	 ○ The participants' preference for interactive call as compared to automated call that can address their personal needs. ○ Since patient B18 (Quote 15) found that the advice on the importance of medication adherence was unnecessary (because he/ she was well aware of that), it makes me think whether IVR intervention really an acceptable mode to provide advice and support for non-adherence? The issue about the type of advice given should be discussed then. 11. Conclusion  ● The article did not include the discussion section.
--	---

REVIEWER	Kristina Curtis Coventry University and Public Health Warwickshire UK
REVIEW RETURNED	20-Jan-2017

GENERAL COMMENTS	This is an interesting and succinct paper exploring patients' and nurses views on non-adherence to medication and acceptance of a IVR. However, you may want to consider including in your limitations section: that more in-depth responses could have been retrieved if interview questions were guided by a comprehensive theoretical framework such as the Capability Opportunity Motivation-Behaviour and Theoretical Domains Framework (Michie, Stralen & West, 2011; Francis, O'Connor & Curran, 2012; Cane, O'Connor & Michie 2012). This would help to explore for example, some of the emotional and social influences on medication adherence. You can refer to Jackson et al. who have applied the COM-B model to medication adherence (Jackson, C Eliasson, L., Barber, N., & Weinman, J. (2011). Applying COM-B to medication adherence. The European Health Psychologist, 7–17). In addition, you could have also used an intervention development framework such as Intervention Mapping or the Behaviour Change Wheel to help map qualitative findings onto theoretical domains and intervention strategies. It would be useful to also include the interview questions in the paper or as a supplementary file.
---

REVIEWER	Dr Rasaan ADISA University of Ibadan, Ibadan, Nigeria
REVIEW RETURNED	28-Mar-2017

GENERAL COMMENTS	The manuscript may be accepted after the authors have address the comments and observations raised. Thank you for the opportunity to review the manuscript titled "Reasons for non-adherence to medication and acceptability of an interactive voice response intervention in patients with hypertension and type 2 diabetes in primary care. A qualitative study" Below are my comments: Overall, the study which I will rather refer to as a pilot study provide
--

a valuable insight and further our understanding of reason for medication non-adherence among hypertensive and T2D patients, as well as tested the acceptability of IVR as an intervention approach to resolve some specific non-adherence problem among patients. However, the following specific comments and observations are worthy of mention

TITLE: I would suggest that the title be rephrased as ----- A qualitative pilot study---- ----- This is because the sample size for the study is too small and therefore can best be suited as a pilot study. Considering the substantial population of patients suffering from hypertension and type 2 diabetes in the UK, likewise in other developed and developing countries. This sample size may not be representative

ABSTRACT

Design, method, participants and setting: For how long was the study take to be completed? i.e. the survey on reasons and the IVR intervention.

As earlier mentioned, this study only considered 19 hypertensive and diabetes patients, and only five practice nurses. How representative of the entire population of interest will this sample be? This is a serious limitation which need to be closely addressed. The more reason why I suggest the title should be rephrased as a pilot study.

Participants presumed to be non-adherent were recruited: How were this ascertained? Is there any preliminary assessment criteria/method to separate adherent from non-adherent?

Results and conclusion: Well written and adequately reflect the study findings.

INTRODUCTION: The introduction is concise and well written with adequate reflection of existing literature to guide the study

METHOD:

Recruitment: Line 9; what does the ----- (name), and in line 11/12----- (place) ----- connote and indicates

Also in line 14 ----- at different levels of deprivation ----- -- To what extent were the participants deprived? Clarification is necessary for reader's understanding. Authors may need to expand further.

Sample size: As reiterated, the sample size is too small and no information was provided on the population of patients with the disease conditions in the study site, so as to guide our understanding of the representativeness of the study sample. This is an important limitation of this study.

RESULTS

Line 24/25: What happen to the remaining 18 participants? (i.e. 19 participants out of 37 patients interviewed). Also, 13 of the 19 patients participated in the follow-up interviews in a few months (How many months specifically?)

Line 25/26: Mean age should be with \pm standard deviation. I also suggest that percent response should be stated with the respective number/frequency for better appreciation of the results. Though, the study is qualitative, it is not out of place to present the results in n

	(%) for better understanding of proportion of patients who have responded in a specific way. Probably the author deliberately avoid such because of smaller sample size. Notwithstanding, it would have been nice if the proportion (i.e. n (%)) is also provided. Table 1: There should be a footnote to explain all the acronyms or alphabet A, B with different subscripts used in the table. The table should be able to stand alone to explain itself. What is DT2? Is it the same thing as T2D. Authors may need to clarify. DISCUSSION: The discussion briefly and appropriately explain the study findings. The discussion can be still be considered appropriate to the study findings, even though it looks more of a pilot study REFERENCES: The references were okay In general the manuscript is well written and further contribute to knowledge of reasons for medication non-adherence among hypertension and type 2 diabetes, while exploring another possible intervention approach (IVR) to resolve non-adherence problems among patients in clinical practice. But there is need for some sentence reconstruction for better comprehension by the reader. The Journal editorial board may therefore wish to considered the paper for publication after the authors have been able to address all the comments and observations raised above. I need to state that, the editor did not make available to me the journal guidelines for author.
--	--

VERSION 1 – AUTHOR RESPONSE

Reviewer 1

Response to general comments

The objectives of this qualitative study was to explore the reasons for non-adherence to prescribed medications, the current actions to address the reported reasons, and test views and recommendations of an IVR intervention to address the reasons of non-adherence to medications. The study aimed to inform the development of an IVR medication adherence intervention for a specific population and at a specific setting and area.

Generally, the aim of this paper was not to explore participants' perception on an IVR intervention, but to inform the development of an IVR intervention. To better report aims and objectives of this study, the objectives at the abstract, and the last paragraph of the introduction have been rephrased.

Obtaining the maximum variation of opinion and views through maximum variation sampling would be a suitable method if the intervention aimed to target a population and settings of a maximum variation (e.g. patients with other than the mentioned health conditions, recruited from different settings, at different regions, different countries etc.), which was not the aim of this study. The manuscript has been amended to better report the aims and objectives.

The reviewer seems to suggest the use of a theoretical sampling framework, where the researchers selectively sample participants based on a pre-defined theoretical framework (e.g. intentional and non-intentional non adherence, levels of adherence). However, this approach of deductive sampling could not satisfy the methodology of intervention development followed by this study. This study was based on the MRC framework for developing and testing complex interventions (see reference 27).

According to this framework, the development of an intervention involves the identification of a theory that best explains the behaviour of interest. To achieve this, the present study selected participants: (a) with the main eligibility criterion being the behaviour of interest (i.e. patients not taking their

medications as prescribed), and (b) within the recruitment setting of interest (i.e. primary care practices located at different deprivation levels), and explored their reasons of their behaviour. This process would enable the identification of a theory/theoretical framework that best explains the behaviour of interests and within the context of interest (i.e. a theory that explains how people can best take their medications as prescribed, between their primary care consultations in Cambridgeshire).

Regarding the triangulation of the findings, although important and could possibly inform the reasons that impact on intervention effectiveness, the purpose of this study was to explore the reasons for medication non-adherence and pre-test the acceptability of the IVR intervention to address these reasons, as an adjunct to primary care. Since this is the first IVR intervention to be developed in this setting (i.e. primary care in the UK), it seemed more sensible to test its acceptability to the targeted population before testing its acceptability to different groups of people (e.g. doctors, pharmacists, family members/carers). The latter would require more resources to be spent, before even knowing what and whether IVR can support medication adherence to the targeted population. As an alternative, we explored these views of others by asking participants about the social influences of their behaviour.

Regarding the design of the study, interviews from patients and health care professionals have been used to obtain multiple perspectives (for more information see: BMJ 2009; 339 doi: <https://doi.org/10.1136/bmj.b4122>).

Response to literature review

- Due to word limitations, it was not possible to provide more information about the IVR and how it works. As an alternative, we have cited two references (see 9, 10) that could provide an overview about the IVR i.e. definition and how it works. However, these references provide information on the different types of IVR delivery modes that have been used in previous interventions. Based on these descriptions we developed the IVR delivery mode, which was further informed by the results of this study (see manuscript: acceptability of the IVR delivery mode).
- The citations 10 and 11 describe the active and modifiable components of the IVR interventions that promote medication adherence changes (i.e. meta-regression of BCTs on intervention effectiveness). Both citations 10 and 11 were informed by the result of this study, which aimed to explore the practices/actions (e.g. simple reminders and/or advice for health behaviour change) to address the reasons for non adherence and further support adherence to medications.
- We have now rephrased parts of the introduction and reported at introduction, last paragraph, first sentence, how this study aimed to address the gap in current practice.

Response to study design

- Sampling of participants is described at page 5, methods, recruitment, fourth paragraph. The process of data collection is described at page 6, interviews.
- The theoretical perspective is described at page 6, first paragraph, first sentence and at page 4, introduction, second paragraph. Data analysis is described at page 7, analysis, first paragraph.

Response to sampling of the participants

- The rationale of the sampling is provided at the response to the general comments.
- The authors described the average list size of the practices within the CCG. The authors did not include the exact number of the list size because this could make the identification of the practice easier and thus affect confidentiality. However, the authors have now included the patient:doctor ratio and patient:staff ratio in table 1, as suggested by the reviewer. Although including this information reveals identifiable information for each practice and could threaten confidentiality.

Regarding the availability of special program. To authors' knowledge, there is no other program available to address non adherence to high blood pressure and/or diabetes type 2 medications. During the interviews, nurses reported that due to time constraints, they usually signpost patients to various websites (e.g. Diabetes UK) to find relevant information. The work burden of the practices is

described at page 11 and reported at quote 10.

- Interviews with the nurses aimed to explore: (a) their view about the reasons for patients' medication non adherence, (b) their actions to support the reasons of medication non adherence, and (c) their views and acceptability of the IVR to support adherence to prescribed medications, as an adjunct to primary care consultations.

As reported at introduction, the need to develop an IVR intervention aimed to fill the gap in current practices to effectively support medication adherence, between primary care consultations. This could not have been achieved without identifying the reasons for medication non-adherence, the current actions within primary care practices to address these reasons, and whether and how the IVR could fit into the current practices to support medication adherence better. The interviews with nurses with experience in advising and supporting patients to adhere to their medications aimed to answer these questions.

- Patients were sampled to explore the reasons of medication non adherence. Since the IVR intervention is a behavioural intervention (i.e. aims to be developed to improve behaviour) it seemed sensible to sample those patients with the behaviour of interest and explore their reasons of the behaviour/non-behaviour. This way, a theoretical explanation of the behaviour/ non-behaviour could be developed.

- Random sampling of eligible patients is relevant to qualitative studies aiming to inform the development of an intervention. For the reasons mentioned at response to general comment, random sampling was selected against theoretical sampling.

- The IVR intervention will be tested to participants with no language barriers. Although it could have been informative, discussion about the impact of interviewing a participant with language barrier could not satisfy the purposes of this study - unless future IVR interventions aim to get translated to a non-English speaking audience.

- Only 13 participants were interviewed using think aloud protocols after pre-testing/ experiencing messages from the IVR intervention and not after they experienced the IVR intervention. The point that data saturation was achieved is described at methods, recruitment, last paragraph.

Response to Theoretical framework/perspective

- This paper had two objectives to facilitate the main aim, which was the development of an intervention.

Objective 1: The identification of the reasons for medication non adherence, and actions to facilitate medication non adherence.

Objective 2: To pre-test IVR messages that included techniques to support medication taking behaviour, Behavior Change Techniques (BCTs).

The objective 1 was informed by a) a review of the current empirical evidence testing what are the theoretical concepts that predict medication adherence, and b) a review of the theories that explain behaviour and behaviour change. The findings from a) and b) were integrated to inform the development of the semi-structure interview guide. To report that better, we included a sentence in the manuscript, methods, recruitment, interviews "The interview guide was informed by previous a review of evidence and theory based research (e.g. 16, 17)". Thus, the semi-structured guide was informed by previous theory and evidence-based concepts. However, the content of these and additional theoretical concepts identified by this study were based on the inductive data analysis (see response to data analysis).

The objective 2 was based on the results of the baseline interviews. The practices/actions reported from patients and nurses were mapped onto the reasons for adherence and non-adherence to medications (i.e. theoretical concepts), and informed the content of the IVR messages. To pre-test the acceptability of these IVR messages this study used a think-aloud protocol. For example, this study pre-tested different strategies targeting different theoretical concepts. Some of these strategies were acceptable and some others were not (see manuscript: results, acceptability of the intervention

content). Those that were acceptable and supportive at promoting adherence will further inform the theoretical framework of the IVR intervention.

- The response about participants' sampling is reported at the general comment. The response about the analysis can be found at the response to data analysis.

Response to data collection

- Data collection period is now reported at page 6, interviews, first paragraph, last sentence and same page, second paragraph, last sentence.
- Semi-structure interview guides are now included as attachment.
- Venues of the interviews. The place of the interviews was selected to be participants' places, so that the influence of the interview context could be noted by the interviewer and integrated into the analysis. For example, at baseline interviews, participants were asked about their actions to facilitate adherence and those actions to address non adherence to medication. Most of the baseline interviews were conducted at participants' houses, and in most cases when participants were asked about their action, they indicated the location they keep their medications and the procedure their follow to take them. During that time they also reported the reasons for adhering or non adhering to their prescriptions.

Similarly to the baseline interviews, the aim for the follow up interviews was to be conducted at participants' places, so that the reason that could impair the delivery (e.g. time between picking up the phone and hearing the message, volume of the voice message) of the intervention could be noted by the researcher. That was perceived to be important for this study, considering that the IVR delivery mode had not been tested before.

Generally, the place of the interviews was selected not only to facilitate comfort and audio recording, but also to identify barriers and facilitators relevant to the intervention delivery mode and content, and further inform the intervention development. To better explain this purpose, we have now included the following sentence at page 6, interviews, third paragraph, and seventh line "to facilitate the collection of contextual information (18)".

- Double checking of the transcription against voice recording was conducted, not only to ensure trustworthiness but also to facilitate analysis of the data (e.g. identifying changes in voices pitch when responding to a question etc). Similarly, field notes were double checked against transcripts to facilitate analysis (e.g. see response to previous comment about venues of the interviews). We have now included this information in manuscript, methods, interviews, third paragraph, last sentence.
- Reflection of the researchers in qualitative research is important, and is implied by the information reported in the manuscript at the page 6, third paragraph, first sentence.

Response to description to the IVR intervention

- The objective 2 of this study was to pre-test the acceptability of the IVR delivery mode and message content, and inform the intervention development; it was not to test the acceptability of the IVR intervention. The aim and objectives at introduction have now been amended to reflect this better. For the acceptability of the delivery mode, this paper described the elements of the delivery mode perceived to influence intervention engagement most, as reported by participants (see manuscript: acceptability of the IVR delivery mode). These were the acceptability of the pre-recorded automatic voice, the navigation options and voice recognition. Supplementary response to this comment can be found at the response to literature review.

For the acceptability of the intervention content (e.g. advice given) a response can be found at the response to Theoretical framework/perspective.

- Messages were pre-recorded using text-to-speech software. The response to your question "how

this method could influence the findings” can be found at the response to the description of the IVR intervention (i.e. element of the delivery mode that could influence intervention engagement).

Response to ethical issues

To make ethical issues more explicit, the following sentences were included in the manuscript:

- Page 5, methods, recruitment, second paragraph, last sentence: Five practice nurses from three practices “provided written informed consent” and took part in the interview.
- Page 5, methods, recruitment, fourth paragraph, last sentence: In total 150 invitations were sent to eligible patients and 19 expressed an interest “and provided written informed consent” to participate in the study.

Response to data analysis

- The objective 1 of this study was to explore the reasons for medication non adherence and the actions taken to address medication non-adherence. In these terms, the interview guide was theory and evidence informed (see: response to theoretical framework/perspectives). The analysis of this study aimed to inform the content of these two concepts “reasons of non-adherence” and “actions taken to address reasons of non adherence”, and identify additional theoretical concepts that could explain medication adherence. To achieve this, the data analysis retrieved inductively the themes and mapped them onto these and additional theoretical concepts (see manuscript: methods, analysis). For example, “not understanding the need to take medications as prescribed” was a theme that was inductively retrieved by the data, to inform “reasons for medication non-adherence”. This process of analysis would enable the development of the theory/ theoretical framework (see response to general comments) of the intervention.
- Period of data collection and analysis is now reported at the manuscript at page 6.
- For the response to this sub-comment see the response to theoretical framework/perspective.

Response to discussion

- The new knowledge produced by this paper was the identification of the reasons of non adherence to medications, as reported by people with hypertension and/or diabetes type 2. To our knowledge reasons of non adherence to medication are currently unknown (see reference 8). This study also pre-tested the acceptability of the IVR to address the reasons of medication non adherence, an adjunct to primary care. To our knowledge, this is the first pre-test of an IVR intervention in this setting.

Not sure how the reviewer concluded that “a number of participants were quite sceptical about developing personalised IVR intervention” since there was no such theme identified and reported in the manuscript. The reviewer could possibly refer to the quote 11, which refers to the acceptability of the automated calls and the text-to-speech software, and not to the personalised content of the intervention. Based on the data from this study, the personalisation of the messages was highly acceptable from participants (see manuscript: results, quote 16).

Regarding the question “Is it really possible to personalise the intervention?”, the response is yes, if you include participant’s name at the messages (see manuscript: results, sentence before quote 16).

To the question “Was there similar intervention that managed to do so?”, the response to this question can be found in studies reporting quantitative meta-analytic evidence (see reference 11).

o That would have been a good question for interviews to assess a RCT at follow up, where intervention group responses (i.e. ivr group) could be compared to the usual care group responses (i.e. control group).

o The quote 15 suggested that messages with information about the duration of the health condition were not acceptable, it does not suggest that the advice about the importance of medication adherence was unnecessary. Providing information about the duration of the health condition and providing information about the health consequences of taking or not taking medications as prescribed are perceived to be different techniques.

The type of the advice given and recommended for future intervention has been amended and

reported in the manuscript, discussion, implications for research and practice.

Reviewer 2

Response to reviewer 2

The authors thank the reviewer for these comments. This study explored the reasons that impacted on participants' capability (e.g. actions to address medication non-adherence), motivation (e.g. beliefs to take medications as prescribed, forgetfulness), and opportunity (e.g. regimens). However, it did not identify all the factors proposed from Jackson et al. 2014, and possibly this could not be achieved by a qualitative study. Mapping all the factors that impact on medication non-adherence explained by COM-B would require data from studies of rigorous design (e.g. cluster RCTs). However, to address this important comment, it was now included in the manuscript, discussion, unanswered questions and future research, last sentence "future studies could explore whether these or additional factors impact on medication adherence (27), using rigorous designs."

Reviewer 3

Response to overall comment

Many thanks for the useful comments.

Response to TITLE:

To authors' knowledge, the sample of qualitative studies does not need to be representative to the whole population with similar conditions, let alone to the population from other developed and developing countries. To authors' knowledge, qualitative studies aim to identify the population of interest and explore the phenomenon under investigation (i.e. medication non adherence) until saturation is achieved. The limitation regarding the generalizability of the study finding is described at discussion "strengths and limitations", last sentence. The authors believe that the more generic title "qualitative study" might include the definition of the suggested methodology. To authors knowledge, the proposed qualitative design is not well defined and thus not suitable for use.

Response to ABSTRACT

Design, method, participants and setting: the time the study taken for the study to be completed is now reported at page 6.

The sample is not representative to the entire population of interest, but it is representative to the population that the intervention would target, if acceptable.

To identify non-adherence patients, practices randomly identified patients who had poorly controlled hypertension and/or type 2 diabetes as indicated by clinical measures (e.g. for hypertension, systolic blood pressure ≥ 140 mmHg/90 mmHg; for T2DM, HbA1c $\geq 6.5\%$); or gaps in ordering their repeat prescriptions (i.e. Cumulative Medication Gap: CMG >0.20 for a period of three months before recruitment). These eligibility criteria were applied to practice databases, where patients are recorded. Eligibility criteria are not reported in the abstract due to word limitation, but they are reported at page 5, third paragraph of the manuscript. The abstract has been amended to better explain recruitment "The study included face-to-face interviews with 19 patients with hypertension and/or type 2 diabetes randomly selected from primary care databases and presumed to be non-adherent"

Response INTRODUCTION

Many thanks for your comments on the introduction.

Response to METHODS

Recruitment: (name) is the name of the CCG. This was included in the manuscript Cambridgeshire

and Peterborough. (place) is the name the area participants were recruited from (i.e. Cambridgeshire and Peterborough). Due to word limitation, it was written CCG in the manuscript. The deprivation of each practice is described in Table 1. The deprivation level of each participant was not explored by this study.

Sample size: Table 1 provides information about the characteristics and the conditions of the participants of this study, which was considered as appropriate for a qualitative study. It was also provided a categorical description of the average size list of each practice within the CCG. However, it seems that the reviewer would be interested to see the number of patients with the health conditions per practice. Eligible practices were those with the highest numbers of patients with type 2 diabetes or hypertension. The exact number of patients per practices is not reported because that would make the identification of the participating practices easier. Alternatively, the number of patients within the CCG is now reported at the footnote of table 1.

Response to RESULTS

Line 24/25 reports “In total 37 interviews were conducted: 19 patients were interviewed at baseline, 13 of whom were interviewed a few months later, and 5 nurses”

This means “in total 37 interviews (i.e. 19 with patients at baseline + 13 with patients at follow up + 5 with nurses = 39 interviews) were conducted: 19 patients were interviewed at baseline, 13 of whom were interviewed few months later (i.e. 19 patients were interviewed at baseline. 13 out of these 19 patients were interviewed few months later), and 5 nurses. The first part of the sentence refers to the total number of interviews and the second to the number of participants.

Line 25/26: The response rate to invitation could be better identified by a systematic review of recruitment strategies.

Table 1. A footnote at Table 1 has now been added to better inform readers.

Response to DISCUSSION

Many thanks for your comments.

VERSION 2 – REVIEW

REVIEWER	Hizlinda Tohid Department of Family Medicine, Faculty of Medicine, Universiti Kebangsaan Malaysia Medical Centre
REVIEW RETURNED	10-May-2017

GENERAL COMMENTS	Thank you for responding to my comments and queries. Changes made based on some of the comments were noted. However, I still feel that most of the issues highlighted previously were not addressed adequately. Thank you for responding to my comments and queries. Changes made based on some of the comments were noted. However, I still feel that most of the issues highlighted previously were not addressed adequately. Among the major issues are related to:  i. the first objective of the study which aimed to explore reasons for non-compliance among the participants. For me, the findings of this objective were superficial and lack rigour due to the study’s methodology, thus it should not be the focus of the manuscript. Nevertheless, I believe the
---

methodology used was appropriate for developing an IVR intervention which was the main aim of the study as mentioned in the introduction. The think aloud protocol was also an appropriate method to assess the acceptability of the IVR intervention. I agree with the author that a study which intended to develop an IVR intervention like this one, baseline interviews to find out patients' reasons for non-compliance, actions to facilitate compliance and opinion about the IVR intervention are important. The findings of these interviews together with literature reviews on previous studies and existing theories can inform the researchers in designing the new intervention. Since the baseline interviews were only a part of the developmental process of the intervention, the methodology had produced superficial and non-holistic findings with regards to reasons for non-compliance. One of the factors that could cause this problem is the study's sampling method, which was random sampling.

In a qualitative study, sampling method that ensure holistic perspectives from people who are involved in the phenomenon of interest is very important. Qualitative researchers need to "purposefully look for variation in the understanding of the phenomenon" (Merriam, 2009). They may need to "purposefully seek data that might disconfirm or challenge their expectations" (Merriam, 2009) and these expectations are usually based on assumptions or pre-existing theories or concepts. Thus, purposeful or purposive sampling is used in a qualitative study and not probability sampling (e.g. random sampling). Lecompte and Preissle (1993) refer this sampling as 'criterion-based selection' whereby qualitative researchers identify a list of essential attributes that potential participants should have. These attributes or criteria must reflect the objectives of the study and those with the attributes are most likely to be the 'information-rich cases' (Merriam, 2009). Interviewing them allows in-depth exploration of the phenomenon of interest. For example, the author did highlight types of non-adherence in the introduction which are based on previous studies. These include intentional and non-intentional non-

adherence. Based on this information, the researchers should purposively sample both types of non-adherent patients. This type of purposeful sampling can also be called as "maximum variation sampling". I disagree with the author's statement: "*Obtaining the maximum variation of opinion and views through maximum variation sampling would be a suitable method if the intervention aimed to target a population and settings of a maximum variation.*" Regardless of the target population of the intervention, if we want to study reasons for non-adherence qualitatively, we still need to ensure that our study capture holistic perspectives regarding reasons for non-adherence. The researchers can even predetermine a number of characteristics/ attributes of interest and not just the type of non-adherence. Due to this, I do not think it is adequate to explore reasons for non-compliance among primary care patients qualitatively. It may be adequate to inform the development of an intervention because other information regarding non-compliance can be obtained from previous studies and existing theories. Unfortunately for me, the findings on reasons for non-compliance were superficial and not holistic enough. So, is it worth to report findings that lack rigour? Appropriate methodology could also produce unique findings as the population understudied may be unique resulting in a generation of new knowledge. However, I feel that this study was unable to produce new knowledge with regards to reasons for non-compliance due to problems mentioned above.

- ii. use of theoretical framework in a qualitative study. In a qualitative study, there are a number of ways how researchers can use a theory or multiple theories. The researchers just need to describe how they use the theory(s) clearly. They could use theory(s) to inform them who they need to sample and it is not considered as 'deductive' sampling like the author highlighted. I'm aware that the IVR intervention was developed based on MRC framework. However, the author should also describe clearly the theory/concept that they used for the qualitative exploration of the phenomenon understudied. For example,

the author did mention that they used behavioural theories to inform the development of the semi-structure interview guide in exploring reasons for non-compliance. So, the author should describe these theories clearly. The interview guide should also be congruent with the theories used i.e. important constructs of the theories should be included in the interview guide and explored during the interviews. Unfortunately, the behavioural theories were not described, therefore it is difficult for readers to decide the credibility of the study with regards to data collection.

I still believe that the author could improve the manuscript by changing its focus. The main aim of the study as written in the introduction was to develop an IVR intervention and assess its acceptability. As highlighted by the author, the 'think aloud protocol' allows exploration of patients' views on "the delivery mode, the content of the messages, and recommendations on how IVR messages could be improved." Presenting these findings should be the focus of the article. The IVR intervention developed by the study could also be presented whereby the content and delivery method can be described clearly. However, for me, the findings on reasons for non-adherence are not worth reported. The methodological problems discussed above can affect the study's validity and reliability in answering its first objective. Due to failure in addressing the major issues, I would not accept the article for publication.

Thank you.

References

1. Merriam SB (2009). Qualitative research: A guide to design and implementation. San Francisco, CA: Jossey-Bass.
2. LeCompte MD, Pressley J (1993). Ethnography and qualitative design in educational research (2nd Ed.). Orland, FL: Academic Press.

REVIEWER	DR Rasaq ADISA UNIVERSITY OF IBADAN, IBADAN, NIGERIA
REVIEW RETURNED	18-May-2017

GENERAL COMMENTS	NO COMMENT. THE PAPER IS WELL WRITTEN
---------------------------------------

VERSION 2 – AUTHOR RESPONSE

Reviewer: 2

There are three topics mentioned at this comment, these are regarding:

- 1) The rationale of the questions.
- 2) The use of a theoretical constructs, BCTs, and possibly the link of these.
- 3) The process to map the interview findings to BCTs.

1) Rationale of the interview questions.

To develop the interview questions for this qualitative study, the process described by Kreuter et al. was followed, which is described in the book: "Tailoring health messages, customizing communication with computer technology, reviewing theories and models".

This process involved the stages described below (in "" is the exact text reported in the book, and below that text is the explanation on the study activities):

a) "What constructs from the theory predict behaviour change":

We conducted a review of intervention studies that tested what theoretical constructs predict medication adherence. For example, the medication adherence trial by Farmer et al., 2012 produced objectively measured medication adherence changes, and suggested that changes in beliefs might have influenced these changes. Thus, one of the interview questions aimed to elicit participants' beliefs about taking medications.

b) "What does theory tell us about behaviours change process?"

We conducted a review of theories that explain behaviours and behaviour change. For example, self-efficacy theory has been suggested to promote behaviour change, thus one of the interview questions aimed to elicit patient's beliefs on their ability to take medications as prescribed, and investigate the different sources that impact on their self-efficacy to take medication as prescribed.

To report this better, the following sentence has been added in the manuscript: "the theoretical concepts that best explained medication adherence change were mapped onto interview questions, and informed the interview schedule. Interview prompts facilitated the exploration of meanings, and new concepts to be emerged".

This process is described in the protocol "What factors influence medication adherence in people with long term conditions and how can these inform an intervention to support adherence to medications in the primary care setting? a longitudinal qualitative research project: a research protocol", which although has been reviewed by ethics, has not been published yet. The protocol has now been referenced in the paper, and is available upon request.

2) The use of a theoretical constructs and/or BCTs.

In my response to your comment I gave an example of how COM-B model could be applied to answer important research questions; and highlighted that this paper could not identify all the constructs proposed by Jackson et al. model to explain the findings. The latter would require data from studies of rigorous design (e.g., RCTs). This has been referenced in the paper, at the discussion, unanswered questions and future research.

3) The process to map the interview findings to BCTs.

The process to map the interview findings to BCTs is described in the protocol: "Developing and pre-testing a tailored interactive voice response (IVR) intervention to support adherence to anti-hypertensive medications: a research protocol" which although has been reviewed by ethics, has not been published yet. The protocol has now been referenced in the paper, and is available upon request.

Many thanks for sending your paper as an example of what you suggest that this paper should be about. However, it seems that your paper described the different stages of the intervention development, and you do not report any information on the acceptability and understanding of the intervention content, or delivery mode, by the population of interest; which suggest that your paper had a different aim and objectives from this paper.

The purpose of this paper was to (a) identify the reasons for non-adherence, and (b) assess the acceptability of the intervention content, including important BCTs, and delivery mode, to address the reported reasons for non-adherence, and possibly medication adherence change.

I agree that it is really important to be transparent in how we develop health behaviour change interventions, to improve their replicability. The process to develop this intervention can be found at the referenced protocol. However, it is also important to explore inductively what the determinants of the behaviour of interest are; whether the techniques to address these determinants are acceptable to patients; and most importantly, to explore patients acceptability and understanding of the techniques/intervention content, and whether they align with the determinants of non-adherence, and the behavior of interest; and this paper focused on describing the latter.

Reviewer: 1

There are three topics mentioned at this comment, these are regarding:

- 1) The first objective of the study
- 2) The sampling method
- 3) The semi-structure interview guide. The answer to this issue can be found at the response to reviewer 2, rationale of the interview questions.

1) The first objective of the study. The first objective of the study was to explore reasons for non-adherence among the participants with type 2 diabetes and/or hypertension. The objective was not to explore the reasons for non-adherence or the holistic perspective of non-adherence. This is particularly important for the development of an intervention. Current guidelines in developing interventions highlight the need to identify- among others- for whom the intervention would work, and in what setting (see West and Michie 2016, the behaviour change intervention ontology). Thus, this study was interested in exploring the reasons for non-adherence in patients with type 2 diabetes and/or hypertension, recruited within the primary care (info reported in the abstract, introduction and methods); and not to identify the reasons for non-adherence. To increase the clarity on the objective one, the title of the article has been amended "Reasons for non-adherence to cardio-metabolic medication".

The aim of the study was not to develop an IVR intervention; which involves different stages, including those mentioned in this paper. However, it seems that the reviewer requires this information to be better explained at the aims of the study; this has now been amended in the manuscript: "this study aimed to develop, and assess the acceptability, of elements of an IVR intervention to support medication adherence to cardio-metabolic medications".

2) Sampling method. In qualitative studies, sampling methods should ensure the collection of the population of interest, and the interview schedule should ensure the collection of in-depth data

regarding the phenomenon of interest.

The sentence you reference "purposefully look for variation in the understanding of the phenomenon" is about the purpose of qualitative research: i.e., to look for variation in the understanding of the phenomenon; this is not about sampling.

Sampling is a method to sample the population of interest, and then, the purpose is to look for variation in their understanding. The sampling method can increase the variation of the population of interest that you sample, not the variation of understandings; and variation of the population can be better achieved by random purposeful sampling, not purposeful sampling. Random purposeful sampling involves the selection of a sample of participants that a) have equal chance of being selected, and b) represent the specific population of interest (e.g., in this case, those with the behaviour of interest). These two elements of the sampling strategy eliminate the sample selection bias.

Random sampling is not based on advanced knowledge of how the outcome would appear (e.g., theoretical sampling); thus knowledge/understanding of the phenomenon can be optimized or produced. In other words, random sampling provided the equal chance for people with the behaviour of interest (i.e., not taking medication), and within different levels of deprivation areas, to be selected. The in-depth interviews allow the exploration of the reasons (or purpose as you mention) that these people do not adhere to their medications, and the acceptability of the intervention content, and delivery mode, to support these reasons. This would allow the test of previous attributes about these reasons, and new attributes to be identified. The latter could not be achieved with pre-defined theoretical sampling, which usually aims to inform the content of a pre-defined theoretical approach.

Regarding applicability of the sampling method: it might be easier to replicate a sampling method that recruits participants from practice databases, using pre-defined and objective inclusion criteria (e.g., age, clinical outcomes); rather than to recruit participants that meet pre-defined and subjectively attributed psychological characteristics to ensure that these attributes are verified. Moreover, current evidence suggest that we do not know, what these attributes are (see reference 8).

Thank you for providing sentences from Lecompte and Preissle (1993) and Merriam (2009). I agree that the sampling criterion should aim to identify a list of essential attributes that potential participants should have. However, for this study the list of essential criteria was the inclusion criteria, with the primary one to be the behaviour or the clinical outcomes. Again, this is a different approach from the sampling approach that you suggest; merely due to the replicability issues reported at the above paragraph.

Regarding your suggestion to purposefully sample patients for intentional and non-intentional non-adherence. I am not sure how do you propose this to be done, but I would be very interested to know. The only way I could think of, would be to have some sort of assessment before interviews, but a) practice databases do not have this sort of information, and b) there is no assessment tool to assess and categorize patients to INA or NINA. Moreover, to authors knowledge, the same patient can have one or both INA and NINA reasons for non-adherence. So, the type of purposeful sampling you suggests, would not be impossible for this research.

However, the response to what you might suggest with "purposeful" sampling, could be found in the manuscript: "interview prompts facilitated the in-depth exploration of meanings, and the identification of themes" and "we decided to cease recruitment when saturation of the data had been achieved". Moreover, your comment might not be about the method to select participants (i.e., randomly identified by practice database), but to the method participants responded to study invitation, which was not random; because only those participants who were interested in taking part in the study, were interviewed. Even though, this was reported at discussion in the limitations "However, consideration should be given when translating these results to different populations (e.g. patients who do not

attend practices or those not participating in research)”; the word “randomly” was removed from the manuscript.

VERSION 3 – REVIEW

REVIEWER	Kristina Curtis Centre for Technology Enabled Health Research (CTEHR) Faculty of Health & Life Sciences Richard Crossman Building (4th Floor) Coventry University Priory Street Coventry CV1 5FB
REVIEW RETURNED	23-Jun-2017

GENERAL COMMENTS	Thank you for addressing all of my comments. I am happy with all the information you have provided.
---